# Evaluating the Effect of Polymer Modification on the Low-Temperature Rheological Properties of Asphalt Binder

**DOI:** 10.3390/polym14132548

**Published:** 2022-06-22

**Authors:** Bagdat Teltayev, Erik Amirbayev, Boris Radovskiy

**Affiliations:** 1Kazakhstan Highway Research Institute, Almaty 050061, Kazakhstan; erik_neo@mail.ru; 2Radnat Consulting, Irvine, CA 90292, USA; b.radovskiy@att.net

**Keywords:** neat and polymer-modified bitumens, low temperatures, creep compliance, relaxation modulus, relaxation time spectrum, glass transition temperature

## Abstract

This paper investigates the viscoelastic properties of oxidized neat bitumen and three polymer-modified binders at low temperatures. The earlier proposed interrelated expressions for the relaxation modulus and for the creep compliance of bitumen binders are further developed. The results of creep testing of the binders on a bending beam rheometer at the six temperatures from −18 °C to −36 °C are presented. The results were analyzed using the equations developed for the relaxation modulus and the relaxation time spectrum. Viscosities at the low temperatures of tested binders were estimated. Approximate interrelations between the loss modulus and the relaxation spectrum were presented. The method for the determination of the glass transition temperature of a binder in terms of the relaxation time spectrum is proposed. The glass transition temperatures of tested binders were determined by the proposed method and compared with ones determined by the standard loss modulus-peak method.

## 1. Introduction

Bitumens are widely used in road paving because of their good adhesion to mineral aggregates and their viscoelastic properties. In paving applications, the bitumen should be resistant to climate and traffic loads, for which reason its rheological properties play a key role. It has to be stiff enough at high temperatures to resist rutting at local pavement temperature around 60 °C while it must remain soft and viscoelastic enough at low temperatures (from −20 °C to −45 °C) to resist thermal cracking. Those requirements are almost opposite, and most of the available neat bitumens would not provide all the needed characteristics together because bitumen is brittle in cold winters and softens readily in hot summers. Moreover, asphalt pavement at intermediate temperatures should be resistant to fatigue cracking from tensile and shear stresses under the action of repeated loading caused by traffic.

In order to enhance neat bitumen properties and widen the service temperature, bitumens are often modified by the addition of polymers. Polymer modification improves mechanical properties, decreases thermal susceptibility and permanent deformation (rutting), and increases resistance to low-temperature cracking. The most commonly used additives are copolymers, such as styrene butadiene styrene (SBS), ethylene vinyl acetate (EVA), styrene- ethylene-butylene-styrene (SEBS). The wide use of this type of polymer for modification is due to its thermoplastic nature at higher temperatures and its ability to form networks upon cooling. Particularly, when SBS is blended with bitumen, the elastomeric phase of the SBS copolymer absorbs the maltenes (oil fractions) from the bitumen and swells up to nine times its initial volume [1]. At SBS concentrations 5–7% by mass, a continuous polymer network (phase) is formed throughout the modified binder, significantly improving the bitumen properties at high and intermediate temperatures.

Until now, the researchers have not developed a convincing opinion on the positive and significant effect of modifying bitumens with polymers at low temperatures. Lu et al. [2] tested three bitumens blended with 6% SBS, EVA, SEBS, or EBA. They concluded that the effect of modification on the low-temperature properties of bituminous binders was small. Authors of [3,4] concluded that modified bitumens have better resistance to low temperature cracking compared to the unmodified ones while Peng [5] found that at −12 °C and −18 °C, the low-temperature stability of the modified bitumen is significantly increased, although at −24 °C it is slightly reduced. Lu et al. in work [6] reported that the glass transition temperature of a bitumen defined as the temperature of the peak loss modulus is reduced by polymer modification; but the results of creep tests performed using a BBR at temperatures of−15 °C, −20 °C, 25 °C, and −30 °C showed that polymer modification does not give beneficial effect; in some cases, especially for the limiting temperature at 0.3 m value, even adverse effect is found for polymer modification. In work [7], dense asphalt concrete samples prepared with bitumens modified with various amounts of SBS polymer were subjected to restrained cooling tests and at a standard cooling rate of 10 °C/h, no significant effect of modification was noted.

Although considerable research was undertaken in this area, the polymer-modified bitumen has still to be comprehensively characterized, due to the complex nature and interaction of the bitumen and polymer system. The present work focuses on the low-temperature rheological properties of the polymer-modified bitumen binders for road pavements.

The low-temperature transverse cracking of asphalt concrete pavements is major pavement distress commonly observed in regions affected by cold weather. Thermal cracking is induced by a rapid drop in temperature that tends to cause contraction and results in tensile stresses that may eventually reach the tensile strength of the material causing its fracture. The field performance data from test roads in Alberta, Manitoba, and Ontario (Canada), and Pennsylvania (USA) indicated that the binder is mostly responsible for the cold-weather cracking of asphalt pavements [8,9,10,11,12]. It became clear that only a fundamental understanding based on sound theory such as binder rheology might provide confidence for moving forward into a low-temperature cracking problem [13,14,15,16,17,18,19,20].

In a pioneering work, Monismith et al. [21] developed a calculation method for the thermally induced stress in the longitudinal direction of asphalt pavement as in an infinite viscoelastic beam. Boltzmann’s superposition principle and the constitutive equation for linear viscoelastic material were applied to relate time-dependent stresses and strains. This approach was then widely used to estimate the development of thermal stresses [12,22,23,24,25,26,27,28]. It turned out that the problem of tensile strength determination for the binder and asphalt concrete was much more complicated, particularly including a ductile-to-brittle failure domain related to the peak to the tensile strength vs. temperature curve [16,24,29,30,31,32].

For many years, researchers have attempted to develop tests that can be incorporated into the binder pass/fail low-temperature specifications. Mandatory binder stiffness (S at 60 s < 300 MPa) and its slope (m value at 60 s, dlogS/dlogt = m > 0.30) at the designed low pavement temperature were included in the standard specifications AASHTO M 320 and ASTM D 6816. Later proposed rheological low-temperature parameters include the difference ΔT between $T_{S=300}$ and $T_{m=0.30}$ [33]; stiffness at m-value = 0.3; Glover parameter that is a combination of storage modulus and a real part of complex viscosity and serves as a surrogate parameter for ductility [34]; Glover–Rowe parameter (G–R) which is the same parameter expressed in other terms [35].

Bitumen contains up to several thousand individual chemical components ranging from non-polar saturated alkanes to polar hetero-hydrocarbons [36]. Phenomenologically, it, therefore, seems natural to view a bitumen as a continuum of molecules with a gradual transition in molecular weight and polarity and with the corresponding continuous spectrum of relaxation times. The change in bitumen binder properties associated with the degree of packing does not occur instantaneously with change in temperature, but requires the passage of time. As the temperature is reduced, then the time scale for these rearrangements increases. In some temperature ranges, this time is of the order of the time scale of the experimental measurement, from minutes to hours. If the temperature is reduced still further, this time scale is increased so much that no further change in order can be observed over practical time scales. The temperature at which this transition occurs is the glass transition temperature.

It was shown that the glass transition temperature of bitumen binders is a reasonable predictor of the temperature close to which the asphalt pavement will thermally fracture [12,37,38]. Since low-temperature cracking is widespread pavement distress, the ability to determine bitumen binders is an important tool for asphalt researchers [18,29,39,40,41,42,43,44,45]. The objective of this paper is to increase the understanding of the low-temperature behavior of bitumen binders, particularly to estimate the glass transition temperature in terms of the relaxation time spectrum.

## 2. Theoretical Background

### 2.1. Linear Viscoelastic Rheological Characterization

To describe the rheological properties of binders we used the earlier proposed model [24,46,47] that includes expressions for tensile relaxation modulus E(t) and for the tensile creep compliance D(t):(1)$$E(t)=E_{g}\;{ [1+\;{\left(\frac{E_{g}t}{3\eta }\right)}^{b}\;]}^{-(1+1/b)},$$
(2)$$D(t)=\frac{1}{E_{g}}{ [1+\;{\left(\frac{E_{g}t}{3\eta }\right)}^{\beta }\;]}^{1/\beta },$$
(3)$$\beta ={ [\frac{1}{b}-\frac{\ln(\pi )}{\ln(2)}+2]}^{-1},$$
where t is a time, s; $E_{g}$ is tensile instantaneous modulus (tensile glassy modulus), Pa; η is the steady-state shear viscosity, Pa·s; b is a constant (0 < b < 1) governing the shape and width of the relaxation time spectrum.

The model [24,46,47] includes expressions (1) and (2) for tensile relaxation modulus E(t) and for the tensile creep compliance D(t). The constant β in Equation (2) was shown to be dependent on the thermal susceptibility of the bitumen binder based on SHELL testing data for 46 bitumens [24]. Equations (2) and (3) were derived by means of linear viscoelasticity in our monograph [46]. Expressions (1) and (2) for tensile relaxation modulus and for the tensile creep compliance match each other (Figure 1).

As it is seen, Equations (1) and (3) include three parameters and all of them have physical meanings.

The relaxation and creep functions E(t) and D(t) are connected by the relation in form of the exact convolution integral [48]:(4)$$C(t)=\int_{0}^{t}E(\xi )\;D(t-\xi )\;d\xi \;=\;t.$$

The convolution product C(t) of Equation (4) was calculated using Equations (1) and (2) at $3\eta /E_{g}=0.01$ s for b = 0.2, 0.3, and 0.4. The results plotted in Figure 1 numerically confirm that the expressions (1) and (2) for relaxation modulus and creep compliance match each other.

The mean relative deviation of the convolution product C(t) from the equality line in Figure 1 is around 1.4% for the time scale of twelve logarithmic decades. Supposing isotropy and incompressibility, Equations (1) and (2) for shear relaxation modulus G(t) and for the shear creep compliance J(t) can be written in the forms:(5)$$G\left(t\right)=G_{g}\;{ [1+\;{\left(\frac{G_{g}t}{\eta }\right)}^{b}\;]}^{-\left(1+1/b\right)},$$
(6)$$J(t)=\frac{1}{G_{g}}{ [1+\;{\left(\frac{G_{g}t}{\eta }\right)}^{\beta }\;]}^{1/\beta },$$
where $G_{g}$ is the shear glassy modulus, MPa.

Complex modulus in shear can be determined from creep compliance Equation (6) using the Schwarzl and Struik method [49]:(7)$$|G^{*}|(\omega )=\frac{G_{g}}{\Gamma (1+m(\omega ))}{ [1+{\left(\frac{G_{g}}{\eta \omega }\right)}^{\beta }]}^{-\frac{1}{\beta }},\delta (\omega )=\frac{\pi }{2}m(\omega ),$$
where $\left|G^{*}(\omega )\right|$ is the norm of complex modulus; δ(ω) is phase angle; ω is the angular frequency, rad/s; Γ(x) is the gamma function, and
(8)$$m(\omega )=\frac{{\left(G_{g}/\eta \omega \right)}^{\beta }}{1+{\left(G_{g}/\eta \omega \right)}^{\beta }}.$$

Here the Van der Poel-Koppelmann [13,50] conversion from time to frequency domain t→1/ω was applied to Equation (6) to derive Equation (7) using [49].

### 2.2. Spectrum of Relaxation Times

According to Bernstein’s theorem [51], every monotonic function can be written as a sum of exponential decay functions exp(−t/τ). The relaxation modulus of a viscoelastic liquid E(t) is a continuous, decreasing function and thus it can be expressed in form of the integral transform [52]:(9)$$E(t)=\int_{0}^{\infty }H(\tau )\exp(-t/\tau )\frac{d\tau }{\tau }=\int_{-\infty }^{\infty }H(\tau )\exp(-t/\tau )d\ln\tau ,$$
where H(τ) is the distribution function of relaxation times τ, shortly the relaxation–time spectrum.

If relaxation modulus E(t) is given, the spectrum H(τ) can be found from the integral Equation (9) by inverting the Laplace transform. Applying the Widder’s inverse Laplace transformation [53] leads to the following asymptotic formula for relaxation–time spectrum:(10)$$H_{n}(\tau )=\frac{{\left(-1\right)}^{n}{\left(n\tau \right)}^{n}}{(n\;-\;1)!}E^{\left(n\right)}\left(n\tau \right),\;E^{\left(n\right)}\left(n\tau \right)={ [\frac{d^{n}E(t)}{dt^{n}}]}_{t=n\tau },$$
where n is the degree of approximation (*n* = 1, 2, 3…).

As n becomes infinite [53], the right side of Equation (10) tends to the exact relaxation–time spectrum H(τ). The convergence rate depends on the relaxation modulus E(t).

Substituting Equation (1) to Equation (10) leads to the following expressions for the first and second approximations of the spectrum:(11)$$H_{1}(\tau )=\frac{E_{g}(1+b){\left(\frac{E_{g}\tau }{3\eta }\right)}^{b}}{{ [1+{\left(\frac{E_{g}\tau }{3\eta }\right)}^{b}]}^{2+1/b}},$$
(12)$$H_{2}(\tau )=\frac{E_{g}(1+b) [(2+b){\left(\frac{2E_{g}\tau }{3\eta }\right)}^{b}+1-b]{\left(\frac{2E_{g}\tau }{3\eta }\right)}^{b}}{{ [1+{\left(\frac{2E_{g}\tau }{3\eta }\right)}^{b}]}^{3+1/b}}.$$

Figure 2 presents an example of calculated spectra for $E_{g}$ = 2.460 × 10^9^ Pa, b = 0.1914, η = 4.247 × 10^6^ Pa·s in the first, second and third approximations. The maximum spectrum density in the first approximation is only 1.6% smaller than in the second and 1.8% smaller than in the third approximation.

Thus, the precision of simple first approximation Equation (11) is acceptable for our purposes. Moreover, Equation (11) has three intriguing features associated with it. First, analytically taking the integral one can obtain exactly
(13)$$\int_{0}^{\infty }\frac{H_{1}(\tau )}{\tau }d\tau =\;E_{g},$$
as it follows from Equation (9) at t=0. The area between the curve $H_{1}(\tau )$ and axis lnτ equals the instantaneous modulus, as it should.

Secondly, analytically taking the integral one can obtain exactly
(14)$$\int_{0}^{\infty }H_{1}(\tau )d\tau =3\eta ,$$
as it should be [48]. The area between the curve $H_{1}(\tau )$ and axis τ equals the elongational viscosity $\eta_{e}=\;3\eta$ [54].

Thirdly, differentiation leads to
(15)$$d\;\log\;H_{1}(\tau )/d\;\log\;\tau =\frac{1+2b}{1+{\left(\frac{E_{g}\tau }{3\eta }\right)}^{b}}-1-b$$

It follows from Equation (15) that at τ=0 the slope $d\;\log\;H_{1}(\tau )/d\;\log\;\tau =b$ while when τ→∞ the slope $d\;\log\;H_{1}(\tau )/d\;\log\;\tau =-(1+b)$. Obviously, the shape parameter b is related to the slopes of the relaxation spectrum. The slope $d\;\log\;H_{1}(\tau )/d\;\log\;\tau =b$ describes the low-relaxation time wing of the spectrum while the slope $d\;\log\;H_{1}(\tau )/d\;\log\;\tau =-(1+b)$ corresponds to the high-relaxation time wing (Figure 3).

Spectrum density $H_{1}(\tau )$ peaks at the modal relaxation time:(16)$$\tau_{m}=\;\frac{3\eta }{E_{g}}\;{\left(\frac{b}{1+b}\right)}^{1/b},$$

## 3. Materials and Methods

### 3.1. Binders

Four asphalt binders were tested. A neat bitumen of penetration grade BND 100/130 produced by direct oxidation from Siberian crude oil by Pavlodar petrochemical plant is commonly used in Kazakhstan paving industry. The second binder was the base bitumen BND 100/130 modified by the reactive ethylene terpolymer Elvaloy 4170 (Du Pont, NY, USA) in the amount of 1.4% by weight. The third binder was the base bitumen modified by the cationic bitumen emulsion of Butanol NS 198 (BASF, Ludwigshafen, Germany) in the amount of 3% by weight. The fourth binder was the base bitumen compounded with a flux (vacuum residue) from the same plant (flow time 82 s at 80 °C) in the amount of 20% by weight and modified by the polymer SBS L 30-01 (Sibur Co., Moscow, Russia)in the amount of 5% by weight.

Elvaloy 4170 is a chemically active copolymer of ethylene (71%) with butyl acrylate (20%) and glycidyl methacrylate (9%). Butanol NS 198 is a cationic, high molecular weight styrene butadiene dispersion designed for use in asphalt modification and waterproofing. The content of solid polymers in Butanol is 64%. SBS L 30-01 A represents a linear block for copolymer of styrene (30%) and butadiene (70%). The molecular weights of the polymers Elvaloy and Butanol are in the range of 60,000–80,000 and 80,000–90,000, respectively. The structural formulas of SBS and Elvaloy polymers are shown in Figure 4.

### 3.2. Preparation of Compounded and Modified Binders

Modification of the bitumen with polymers Elvaloy and Butanol was carried out in accordance with the normative documents of Kazakhstan [55] and [56], respectively. The polymers Elvaloy and Butanol were gradually added to the heated neat bitumen at 170 °C using a laboratory mixing device. Continuous mixing process of polymer–bitumen binders lasted for two hours, the next twelve hours the Elvaloy modified binder was conditioned at constant temperature of 170 °C.

Compounding of the base bitumen with the flux was performed by means of mixing with the rate of 450–500 rotations per minute at the constant temperature of 120 °C for 30 min [57]. Then, the compounded bitumen was gradually heated up to 180 °C and polymer SBS was gradually added. During the first two hours and next four hours, a mixing rate was equal to 1200 and 1800 rotations per minute, respectively.

### 3.3. Conventional Properties of Binders

Conventional properties of the binders were defined in the Research Laboratory of Kazakhstan Highway Research Institute according to the test specification ST PK 1373−2013 and they are presented in Table 1.

### 3.4. Rheological Testing

The binders were tested at low temperatures (−18 °C, −24 °C, −27 °C, −30 °C, −33 °C, and −36 °C) on the ATS Bending Beam Rheometer (BBR) according to the standard ASTM D 6648.

Prior to the rheological testing, the binders were aged using a Rolling Thin-Film Oven (RTFO) according to the test specification ASTM D2872 to simulate the short-term aging during asphalt mix manufacture. Then the binders were further aged in the Pressurized Aging Vessel (PAV) to simulate the long-term aging (ASTM D6521).

The binder samples for the testing had a shape of a beam with dimensions 6.25 × 12.5 × 125 mm. The duration of specimen conditioning prior to the testing was set to one hour. In the BBR creep test a constant load *P* = 0.98 N was applied at the midpoint of the simply supported binder beam for 240 s. The mid-span deflection d(t) was constantly recorded. The creep stiffness S(t) was automatically calculated from the equation:(17)$$S(t)=\;\frac{PL^{3}}{4wh^{3}\;d(t)},$$
where L is the span of the beam (102 mm); w is the width of the beam (12.5 mm); h is the height of the beam (6.25 mm); d(t) is maximum deflection of the beam at time t.

Only the observations at 8, 15, 30, 60, 120, and 240 s were employed in the present study.

## 4. Results and Discussion

### 4.1. Stiffness and Viscosity

As an example, the results of testing are shown in Figure 5 for the Elvaloy modified bitumen binder.

To produce a master curve at a selected reference temperature $T_{r}$, Equation (2) combined with the Arrhenius time-temperature superposition function was used:(18)$$S(t)=\frac{1}{D(t)}=E_{g}{ [1+\;{\left(\frac{E_{g}t}{3\eta }\right)}^{\beta }\;]}^{-1/\beta },$$
(19)$$\eta =\eta_{r}\exp\; [\frac{\Delta H_{a}}{R}\;\left(\frac{1}{273+T}-\frac{1}{273+T_{r}}\right)],$$
where $\eta_{r}$ is a viscosity at a reference temperature, Pa·s; $\Delta H_{a}$ is the flow activation energy, J/mol; *R* is the universal gas constant equal to 8.314 J/(mol·K).

The parameter β is related to b as before by Equation (3).

Based on our previous study [24,46], the instantaneous tensile modulus was assumed $E_{g}$ = 2460 MPa for all tested binders. The reference temperature was selected close to the midrange of testing temperatures $T_{r}$ = −30 °C. Using the Mathcad software, a nonlinear minimization algorithm was implemented to determine simultaneously the parameters $\eta_{r}$, $\Delta H_{a}$, and b by minimizing the sum of squared deviations of data points from the master curve S(t) Figure 6.

The obtained values of the parameters are given in Table 2. Figure 7 shows the viscosity as a function of temperature calculated using Equation (19).

The slopes of the viscosity-temperature relationships to the temperature axis for the binders modified by the polymers Elvaloy and Butanol are almost equal and they are smaller than the slopes for the neat bitumen and the bitumen compounded by the flux and modified by the polymer SBS. This indicates the lower temperature susceptibility of the Elvaloy and Butanol-modified binders. In the range of the testing temperatures, the Elvaloy-modified binder has the smallest viscosity while the Butanol-modified binder (at temperatures higher than −27 °C) and the flux compounded and SBS-modified −27 °C) have the greatest one. The ability to estimate the viscosity of a binder at subzero temperatures indirectly from conventional BBR testing is a useful feature of the paper.

### 4.2. Glass Transition Temperature in Terms of Loss Modulus

Several researchers have shown that the glass transition temperature $T_{g}$ of a bitumen binder is associated with the low-temperature cracking of a pavement [12,37,38]. The transition to a glassy state increases the brittleness of the binder, reducing its ability for stress relaxation and resulting in higher cracking susceptibility of an asphalt pavement. Researchers measured the glass transition temperature of bitumen binders by using three different techniques: dilatometry, calorimetry and rheological method-peak in the loss modulus versus temperature.

The classic method for the determination of the glass transition temperature is dilatometry. The temperature dependence upon cooling of the specific volume is determined by a suitable technique, and the temperature at the change in slope is taken as $T_{g}$ at a given cooling rate [8,39,40,41]. Because of the need for precise measurements of small changes in volume with decreasing temperature, the dilatometric method is a difficult procedure to perform. Calorimetry was extensively employed, a peak in heat capacity being observed at the temperature $T_{g}$, depending on the heating rate [29,42,43].

Last year, the rheological dynamic measurements were conducted to estimate the glass transition temperature of bitumen binders. The data are collected over the temperature range at constant frequency and the loss modulus G″ (or E″) peak temperature is taken as $T_{g}$. The standard ASTM 1640 [58] recommends the testing frequency 1 Hz (ω = 6.28 rad/s). This standard admits other frequencies but they should be reported. Anderson and Marasteanu [41] used frequency 0.1 rad/s, Reinke and Engber [44] used 0.1–1.0 rad/s, Planche et al. [43] used 5 rad/s, Sun et al. used 10 rad/s [45]. Changing the time scale by a factor of 10 will generally result in a shift of about 8 °C for a typical amorphous material [58]. Anderson and Marasteanu [59] compared dilatometry, calorimetry, and the peak in the G” and concluded that all methods give estimates of the glass transition temperature that are in relatively good agreement, given the different nature of the measurements and the time scale of the measurements.

In the present study, $T_{g}$ is defined based on BBR testing as the temperature where the tensile loss modulus $E^{\prime \prime }$ peaks at frequency 1 rad/s. The tensile loss modulus $E^{\prime \prime }$ can be determined from tensile creep compliance D(t) using the Schwarzl and Struik method [49] from the equation
(20)$$\begin{array}{cc} & E^{\prime \prime }(\omega )=\frac{E_{g}\sin\left(\frac{\pi }{2}m_{E}(\omega )\right)}{\Gamma (1+m_{E}(\omega ))}{ [1+{\left(\frac{E_{g}}{3\eta \omega }\right)}^{\beta }]}^{-\frac{1}{\beta }},\\ \mathrm{where} & \\ & m_{E}(\omega )=\frac{{\left(E_{g}/3\eta \omega \right)}^{\beta }}{1+{\left(E_{g}/3\eta \omega \right)}^{\beta }}. \end{array}$$

Loss modulus $E^{\prime \prime }(\omega )$ was calculated from Equation (20) using Equation (19) and the parameters are shown in Table 2 at the frequency ω = 1 rad/s. When the loss modulus $E^{\prime \prime }$ is plotted versus temperature, the resulting curve exhibits a peak value (Figure 8). The temperature at this peak value can be interpreted as a glass transition temperature $T_{g}.$ The temperatures at which the calculated $E^{\prime \prime }$ reached a maximum value were $T_{g}$ = −45.4 °C for the neat bitumen, $T_{g}$ = −52 °C for the Elvaloy-modified binder, $T_{g}$ = −46.3 °C for the Butanol-modified binder and $T_{g}$ = −46.2 °C for the flux-compounded and SBS-modified binder. Changing the “testing” frequency by a factor of 10 results in a $T_{g}$ shift of 5.5–7 °C.

At low polymer concentration (not more than about 5%), the properties of the base bitumen and the compatibility of polymer with bitumen are very important. Modification with 3% Butanol primarily aims to improve the high-temperature properties of the bitumen and does not show a positive effect at subzero temperatures compared with the base neat bitumen in this study. Considerable improvement of high-temperature properties (softening point from 45 °C up to 76.5 °C) of the base bitumen, keeping its low-temperature properties, was achieved by means of compounding by flux and modifying by polymer SBS. Modification with 1.4% Elvaloy, which is intended to enhance the high-temperature properties as well, also improved the low-temperature properties. Modification with Elvaloy lowered the temperature susceptibility, lowered the low-temperature viscosity and reduced the glass transition temperature. Particularly, these results are important for countries with a sharp-continental climate, including Kazakhstan, which is the ninth-largest in the world where half of the territory requires the bitumen binder of Superpave grade PG 58−40.

### 4.3. Glass Transition Temperature in Terms of Relaxation Time Spectrum

#### 4.3.1. Interrelations between Loss Modulus and Relaxation Spectrum

To increase the understanding of the glass transition temperature of the binder it would be of interest to express $T_{g}$ in terms of the relaxation time spectrum. In the previous section, the widely accepted definition of the glass transition temperature was used, which consisted of taking the loss modulus (G″) maximum as a function of temperature at a given frequency ω. Loss modulus is related to the relaxation–time spectrum by an exact equation [48].
(21)$$G^{\prime \prime }(\omega )=\int_{0}^{\infty }\frac{H(\tau )}{\tau }\cdot \frac{\omega \tau }{1+\omega^{2}\tau^{2}}d\tau .$$

The kernel of the loss modulus function
(22)$$f(\tau )=\frac{\omega \tau }{1+\omega^{2}\tau^{2}},$$
is a crude approximation to the Dirac delta function $\delta_{D}(\tau )$. The function f(τ) reaches a maximum at τ=1/ω. Since the integral
$$\int_{0}^{\infty }\frac{\omega \tau }{1+\omega^{2}\tau^{2}}d\tau =\frac{\pi }{2}$$
is not unity as it is for the delta function $\delta_{D}(\tau )$, the function f(τ) must be multiplied by 2/π before being replaced by the delta function. Then
(23)$$\frac{2}{\pi }\cdot \frac{\omega \tau }{1+\omega^{2}\tau^{2}}\approx \delta (1-1/\omega ).$$

If Equation (23) is substituted into Equation (21), and using the sifting property of the Dirac delta function yields the relation:(24)$$G^{\prime \prime }(\omega )\approx \frac{\pi }{2}H\left(\frac{1}{\omega }\right).$$

Applying the Van der Poel-Koppelmann conversion from the frequency domain to the time domain ω→1/τ leads to the equation:(25)$$H(\tau )\approx \frac{2}{\pi }{\left.G^{\prime \prime }(\omega )\right|}_{\omega \to 1/\tau }.$$

It follows from Equation (25) that a relaxation time spectrum H(τ) is approximately proportional to a loss modulus function $G^{\prime \prime }(\omega )$. Their shapes are similar and their maxima represent the concentration of the relaxation time spectrum or correspond to the relaxation frequency spectrum in a certain region of the logarithmic τ or ω scales.

#### 4.3.2. Glass Transition Temperature

Calculated spectrum densities at the modal relaxation time *τ_m_* = 1/*ω_m_* = 1 s using Equation (11) and the parameters given in Table 2 are presented in Figure 9. The curves in Figure 8 and Figure 9 look very similar with allowance for approximations in the derivation of Equations (11) and (25).

For a given modal frequency, e.g., *ω_m_* = 1 rad/s, loss modulus $G^{\prime \prime }(\omega )$ at a certain temperature has a maximum, e.g., $T_{g}$ = −45.4 °C for bitumen (Figure 8). According to time–temperature superposition principle, it equivalently means that for the constant temperature −45.4 °C the loss modulus $G^{\prime \prime }(\omega )$ has a maximum at the modal frequency *ω_m_* = 1 rad/s. Similarly, in the relaxation time domain, the relaxation time spectrum H(τ) has a maximum at the modal relaxation time *τ_m_* = 1/*ω_m_* [Equation (16)].

Substituting Equation (19) into Equation (16) leads to the relation:(26)$$\tau_{m}=\frac{3\eta_{r}}{E_{g}}{\left(\frac{b}{1+b}\right)}^{1/b}\exp [\frac{\Delta H_{a}}{R}\;\left(\frac{1}{273+T}-\frac{1}{273+T_{r}}\right)],$$
where the modal relaxation time $\tau_{m}$ corresponds to the arbitrary temperature T.

The ratio of the modal relaxation time $\tau_{mg}$ at a temperature $T_{g}$ to the modal relaxation time $\tau_{mr}$ at a reference temperature $T_{r}$ is
(27)$$\frac{\tau_{mg}}{\tau_{mr}}=\exp [\frac{\Delta H_{a}}{R}\;\left(\frac{1}{273+T_{g}}-\frac{1}{273+T_{r}}\right)],$$

It follows from (27) that the fixed $\tau_{mg}$ glass transition temperature $T_{g}$ can be calculated by equation:(28)$$T_{g}={ [\frac{1}{273+T_{r}}+\frac{R}{\Delta H_{a}}\ln\left(\frac{\tau_{mg}}{\tau_{mr}}\right)]}^{-1}-273,$$
where $\tau_{mr}$ is the modal relaxation time at a reference temperature *T_r_*:(29)$$\tau_{mr}=\frac{3\eta_{r}}{E_{g}}{\left(\frac{b}{1+b}\right)}^{1/b},$$

For the fixed $\tau_{mg}$ = 1 s, using the parameters shown in Table 2, the calculated from Equation (28) glass transition temperature equals $T_{g}$ = −45.8 °C for bitumen, $T_{g}$ = −52.4 °C for the Elvaloy-modified binder and $T_{g}$ = −46.7 °C for the Butanol-modified binder, which almost coincides with $T_{g}$ determined from the peak of loss modulus at frequency ω = 1 rad/s (Figure 8). The calculated dependence of glass transition temperature on modal relaxation time is presented for the tested binders in Figure 10.

The ability to estimate the glass transition temperature in terms of relaxation time spectrum is important for understanding the behavior of binders at low temperatures. The molecular mobility of liquids including bitumen binders expresses itself in a relaxation–time spectrum. The broader the molecular weight distribution is the broader the relaxation spectrum becomes [60,61]. When a liquid-cooled down, its volume reduces due to the translational molecular readjustments rather than due to their oscillating motions. When a temperature reaches the glass transition region, the speed of molecular adjustment becomes slower and no further change in order can be observed over a given time scale. Physically, the glass transition occurs at the temperature $T_{g}$ when the root-mean-squared displacement of the particle of average molecular weight becomes smaller than the average size of that particle [62], on a given relaxation timescale (of the order of τ = 1 s, for example). Relaxation at a temperature lower than $T_{g}$ occurs mostly due to the oscillating motion of molecules. Thus, the glass transition is the transformation of a disordered state with molecular mobility to an immobilized state of a similar structure by means of decreasing temperature. The transformation of liquid (e.g., a bitumen binder) is caused by a continuous increase in the modal relaxation time up to the given scaling time. It can be the time scale of practical observation or an experiment. Relating the bitumen binder properties or performance to the particular scaling time value requires more research.

It is known that the addition of a polymer to a bitumen increases its viscosity, and one would expect a significant decrease in its low-temperature characteristics. However, the results of this work showed that when the bitumen is modified with the given amounts of selected polymers, a positive effect can be obtained. This can be explained by the lower glass transition temperatures of the elastic parts of the polymers (butyl acrylate in Elvaloy, butadiene in SBS and Butanol).

## 5. Conclusions

The main conclusions and findings based on the analysis presented in this paper are as follows:An approximate interrelation between a loss modulus and a relaxation–time spectrum was presented.The glass transition temperature of binders was calculated by the presented rheological method in terms of the relaxation–time spectrum. Calculations by this method showed that: (1) modification of bitumen with 3% Butanol NS 198 that primarily aims to improve the high-temperature properties does not show a positive effect at subzero temperatures compared with the base pure bitumen; modification of bitumen with 1.4% Elvaloy 4170, which is primarily intended to enhance the high-temperature properties, also improved the low-temperature properties; modification with Elvaloy lowered the temperature susceptibility of binder, lowered the low-temperature viscosity and lowered the glass transition temperature by about six degrees (from −45.4 °C to −52.0 °C).The proposed model for stiffness modulus (Equation (1)) enables the estimation of the viscosity of the binder at low temperature indirectly from conventional BBR testing. Viscosity values at temperature −30 °C were 3.520 × 10^6^ MPa∙s for the oxidized bitumen of penetration grade 100/130, 1.374 × 10^6^ MPa∙s for the bitumen modified by the polymer Elvaloy, 6.411 × 10^6^ MPa∙s for the bitumen modified by the polymer Butanol and 8.143 × 10^6^ MPa∙s for the bitumen compounded by flux and modified by the polymer SBS. The temperature susceptibility of the Elvaloy- and Butanol-modified binders was lower than for the neat bitumen and the flux compounded and SBS modified bitumen.In order to determine the optimal content of polymers in bitumens, it is recommended to continue the study of their rheological and other characteristics in the future by varying the technological conditions of modification.

## Figures and Tables

**Figure 1 polymers-14-02548-f001:**
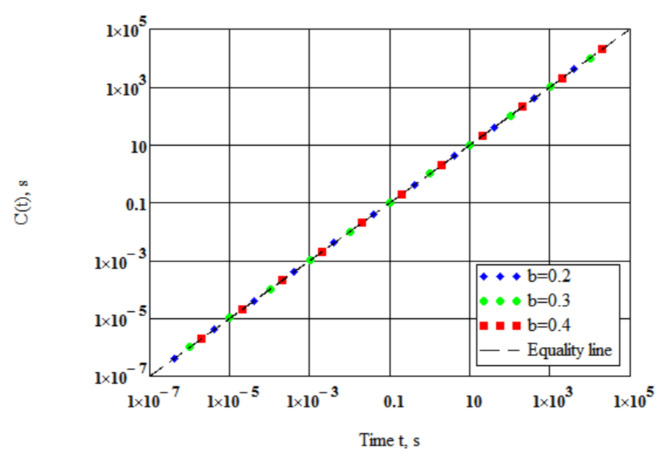
The convolution product of relaxation modulus and creep compliance.

**Figure 2 polymers-14-02548-f002:**
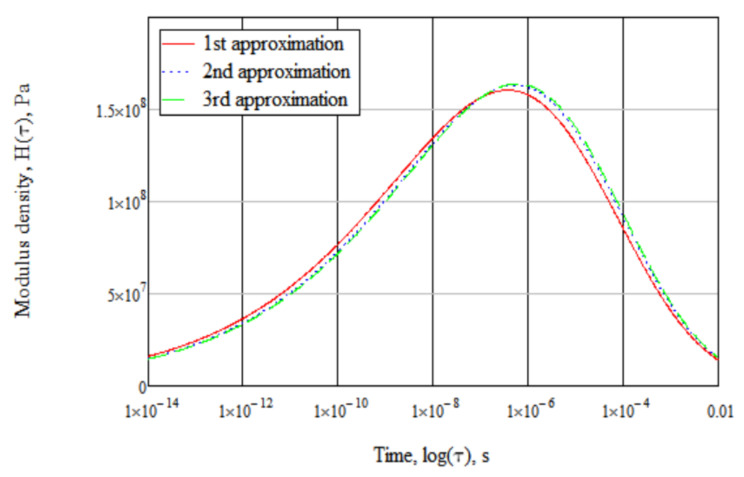
Comparison of calculated spectra.

**Figure 3 polymers-14-02548-f003:**
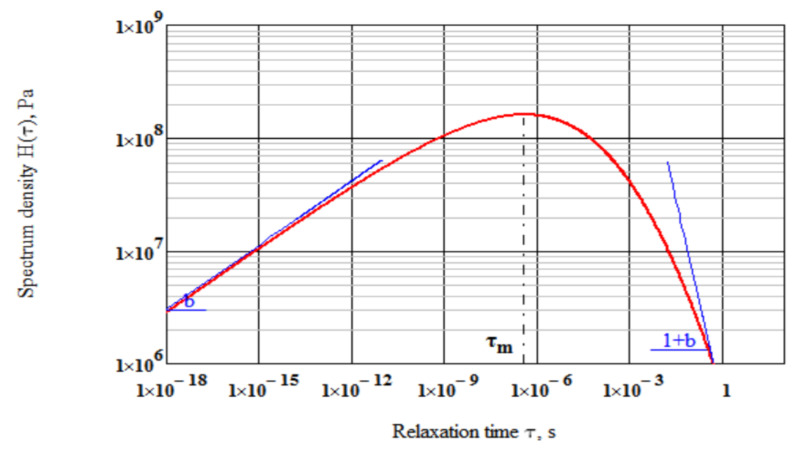
Spectrum and geometrical meaning of parameter b.

**Figure 4 polymers-14-02548-f004:**
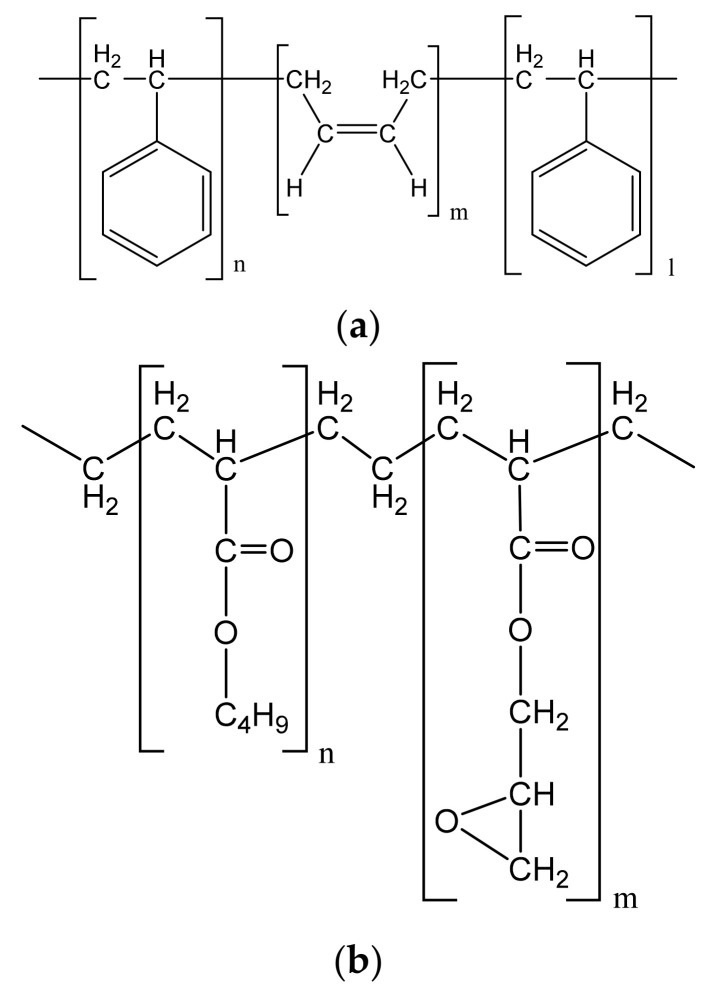
The structural formulas of (**a**) SBS and (**b**) Elvaloy polymers.

**Figure 5 polymers-14-02548-f005:**
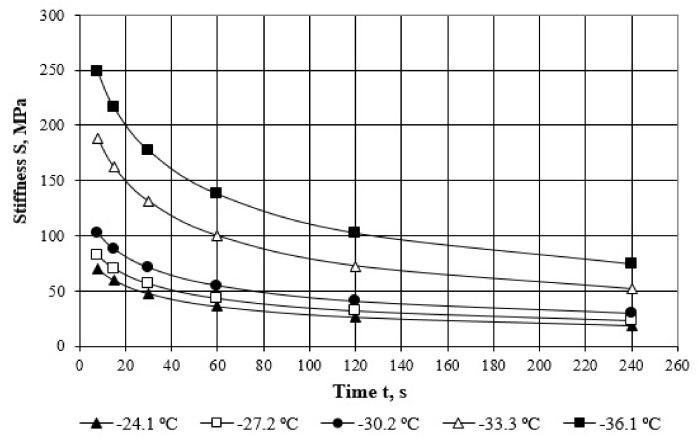
Time-dependent stiffness at different temperatures for Elvaloy modified binder.

**Figure 6 polymers-14-02548-f006:**
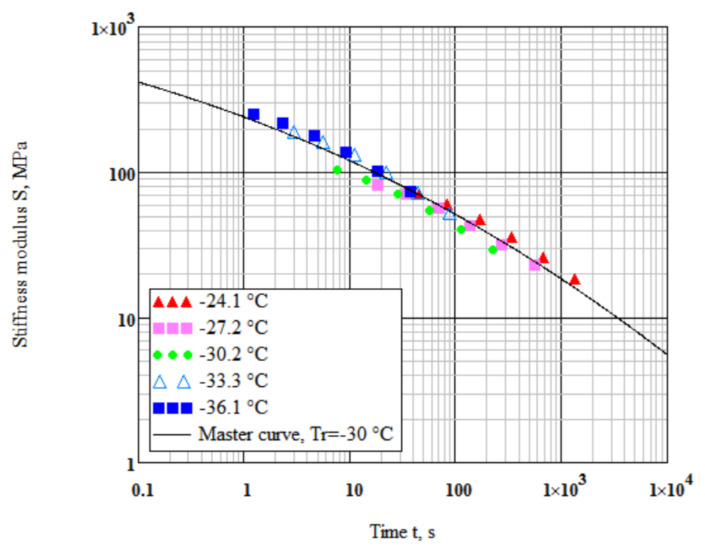
Master curve of stiffness as a function of time for the Elvaloy modified bitumen binder at $T_{r}$ = −30 °C.

**Figure 7 polymers-14-02548-f007:**
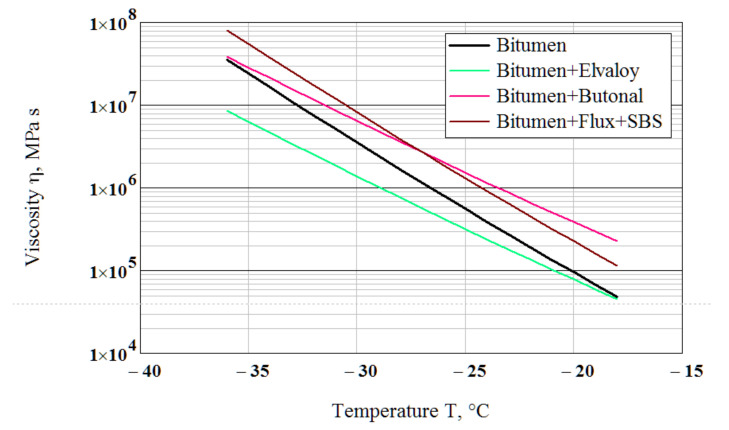
Viscosity–temperature relationships for the binders.

**Figure 8 polymers-14-02548-f008:**
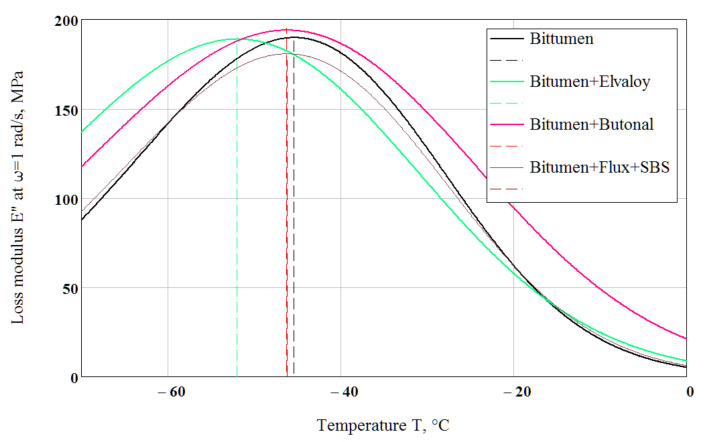
Calculation of the glass transition temperatures for the tested binders using the loss modulus peak method at frequency ω = 1 rad/s.

**Figure 9 polymers-14-02548-f009:**
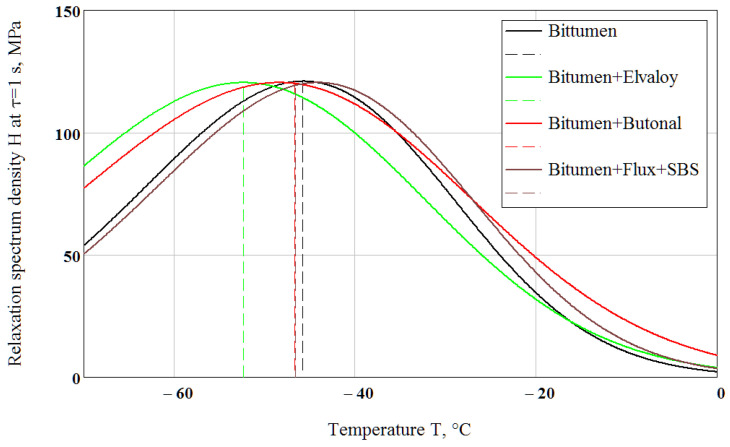
Definition of the glass transition temperatures for the binders using relaxation spectrum density at the modal relaxation time *τ_m_* = 1 s.

**Figure 10 polymers-14-02548-f010:**
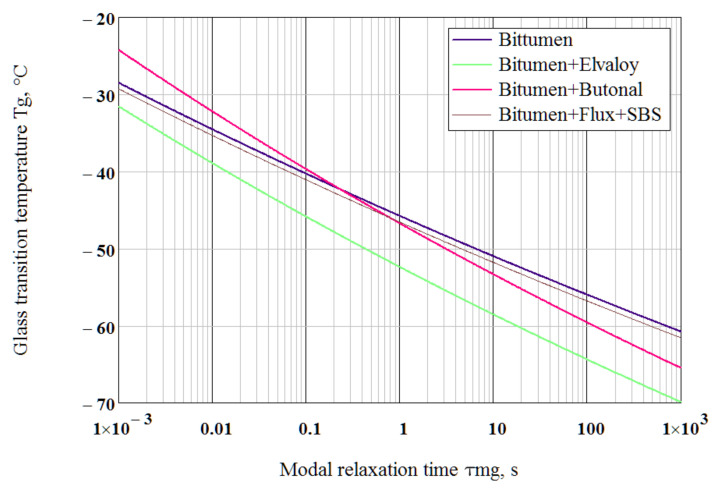
Dependence of glass transition temperature on modal relaxation time.

**Table 1 polymers-14-02548-t001:** Conventional properties of binders.

Property	Standard	Bitumen	Bitumen Modified by		
			Elvaloy	Butanol	Flux + SBS
Penetration, 25 °C (0.1 mm)	ASTM D5	103	86	88	73
Penetration index PI	EN 12591	−0.70	-	-	-
Ductility (cm), 25 °C (cm)	ASTM D113	>150	32	63	70
Softening point (°C)	ASTM D36	45.0	62.5	62.0	76.5
Fraass breaking point (°C)	EN 12593	−26.4	−25.8	−25.4	−23.0
Flash point (°C)	ASTM D92	265	250	265	-
Dynamic viscosity, 60 °C (Pa·s)	ASTM D2171	167	-	-	-
Kinematic viscosity, 135 °C (mm^2^/s)	ASTM D445	394	-	-	-
Elastic recovery, 25 °C (%)	ASTM D6084	-	87	81	98

**Table 2 polymers-14-02548-t002:** Values of the parameters for binders.

Binder	η at $T_{r}$ = −30 °C, MPa∙s	$\Delta H_{a}$, J/mol	b
Bitumen	3.520 × 10^6^	1.847 × 10^5^	0.1418
Bitumen + Elvaloy	1.374 × 10^6^	1.468 × 10^5^	0.1412
Bitumen + Butanol	6.411 × 10^6^	1.437 × 10^5^	0.1452
Bitumen + Flux + SBS	8.143 × 10^6^	1.833 × 10^5^	0.1346

## Data Availability

Not applicable.

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
