# Peer review of "Evaluating the Effect of Polymer Modification on the Low-Temperature Rheological Properties of Asphalt Binder"

_polymers, 2022, doi:10.3390/polym14132548_

Round 1
Reviewer 1 Report
The article by Teltayev B. et al. describes measuring the rheological characteristics of four bitumens at low temperatures by bending and measuring strain over time. The authors give a detailed theoretical background, measure the time dependence of stiffness at several temperatures, use the principle of temperature-time superposition to reduce the data to a single temperature and then recalculate them to obtain temperature dependences of the loss modulus, generate a relaxation time spectrum and estimate the glass transition temperature of the samples. In general, the article is original and clearly written, but its problem is a poor connection to polymer science, given the subject matter of the journal to which it is submitted – Polymers. Yes, the authors examine three samples of bitumen, which contains polymer additives. However, it is not enough just to test samples containing a polymer, it is necessary to give scientific interpretations related to polymers, at least why the polymeric nature of additives is important. In my opinion, the article can be published, but it is highly desirable that the authors would strengthen the polymer part if they want to publish it exactly in Polymers.
Specific comments on the article are as follows.
Lines 37-39: “After removing unnecessary coefficient 3.0 in the Equation (21) [14] because the pavement thermal contraction restrained only in longitudinal direction, calculations give the reasonable values of tensile stresses induced by cooling ([21] Discussions).” This sentence is really unclear. What equation? Why 21? Where do I look for it? In the discussion? If the authors want to cite an equation here, they should do so explicitly with appropriate explanations.
Line 69-71. It is not clear from the purpose of the paper, as well as from the entire introduction, how the topic of the authors' article is related to polymers. I think the authors should explain in the introduction how their topic relates to polymer science or submit their paper to another journal, such as Materials.
Line 156: “reactive ethylene terpolymer Elvaloy 4170”. Can the authors give more information? What is the structure of the three monomers? What is their ratio? What are the reaction groups of this polymer and how are they arranged? What is its molecular weight? The same is true for Butonal NS 198 and SBS L 30-01. What are the emulsion concentration and the structure and molecular weight of the polymer? What are the molecular weight and styrene content of SBS? It is highly desirable to give as much information as possible about the modifiers used.
Line 205: “Arrhenius time-temperature superposition function were used”. The Arrhenius equation usually works very poorly at temperatures close to the glass transition temperature (the activation energy becomes temperature dependent). Why did the authors not use the Williams-Landel-Ferry equation? It would have been much more accurate and correct.
Table 2. MPa·s -> Pa·s? Is there no error here? The same applies to the Y-axis in Figure 6.
Lines 245-246, 254-256: “method - peak in the loss modulus versus temperature.” The glass transition temperature of bitumens and other heavy hydrocarbons can also be determined using the WLF equation using the temperature dependence of their viscosity (see, e.g., doi 10.1021/acs.energyfuels.7b03058, 10.1016/j.petrol.2021.108641). The use of the maximum loss modulus position is not the only rheological method for determination of glass transition temperature.
Lines 380-383: “Viscosity values at temperature -30°Ð¡ were…” What is the value of angular frequency or shear rate? Bitumens are strongly non-Newtonian systems at this low temperature.
Author Response
Responses to the Reviewers' comments are attached.

Reviewer 2 Report
Evaluating the effect of polymer-modification on the low temperature rheological properties of asphalt binder. Topic is under the scope of polymer journal, however, this is not new topic. Thus, authors must show why this paper should be accepted. authors need to show the novelty of this paper.
abstract: line 15-18"The method for determination of the glass transition temperature of a binder in terms of the relaxation time spectrum is proposed. The glass transition temperatures of tested binders were determined by the proposed method and compared with ones determined by the standard loss modulus-peak method" ,,,, authors should show specific data of the findings.
- Introduction, this is the most important section and authors show in sufficient research to cover the objective of the current study. Thus, improving the introduction is must.
section 3.2: Preparation of compounded and modified binders .Why the authors select these conditions of temperatures, time and rpm? Please clarify and support it with references.
line 75: "To describe the rheological properties of binders we used the earlier proposed model [17, 39, 40], please give more information about the model by [17, 39, 40].
figures 1-3, please give Ref.?
conclusion:
give data and strong conclusion about " Modification of bitumen with 1.4% Elvaloy 4170"...
And clarify the optimum binder to be the ideal binder.
finally, how this study ranked the low-temperatures improvement based on the methods of this study. And what will be future works/ recommendations.
Author Response

(The authors gave the same response as above.)

Round 2
Reviewer 1 Report
The authors have made all the recommended corrections, so their article can be published in Polymers.
Reviewer 2 Report
Thanks for the author’s reply and. Improvement of the paper.